# Defining the nutritional input for genome-scale metabolic models: A roadmap

**Georgios Marinos** [1], **Christoph Kaleta** [1]*, **Silvio Waschina** [2]*

**1** Research Group Medical Systems Biology, Institute of Experimental Medicine, Kiel University, University Medical Center Schleswig-Holstein, Kiel, Schleswig-Holstein, Germany, **2** Division of Nutriinformatics, Institute for Human Nutrition and Food Sciences, Kiel University, Kiel, Schleswig-Holstein, Germany

* s.waschina@nutrinf.uni-kiel.de (SW); c.kaleta@iem.uni-kiel.de (CK)

## Abstract

The reconstruction and application of genome-scale metabolic network models is a central topic in the field of systems biology with numerous applications in biotechnology, ecology, and medicine. However, there is no agreed upon standard for the definition of the nutritional environment for these models. The objective of this article is to provide a guideline and a clear paradigm on how to translate nutritional information into an *in-silico* representation of the chemical environment. Step-by-step procedures explain how to characterise and categorise the nutritional input and to successfully apply it to constraint-based metabolic models. In parallel, we illustrate the proposed procedure with a case study of the growth of *Escherichia coli* in a complex nutritional medium and show that an accurate representation of the medium is crucial for physiological predictions. The proposed framework will assist researchers to expand their existing metabolic models of their microbial systems of interest with detailed representations of the nutritional environment, which allows more accurate and reproducible predictions of microbial metabolic processes.

**Data Availability Statement:** Supplementary Data and the R scripts to replicate the results are deposited in GitHub (www.github.com/maringos/StepDiet) (version / commit: 3385525).

## Introduction

A key approach in Systems Biology is the elucidation of the metabolic potential of an organism and the transformation of this information into networks of metabolic reactions. Such networks are called genome-scale metabolic models and can be used in computer simulations of metabolism of cells and cellular communities. For instance, models for various eukaryotic and prokaryotic organisms and cell types have been reconstructed (e.g. as reviewed in [1]) and applied in microbial ecology, biotechnology, and medical research [2–7].

By mathematical approaches, such as the application of Flux Balance Analysis (FBA), we are able to optimise a predefined objective function, which is based on nutritional constraints that represent the chemical environment of cells [4]. A widely used objective function is the production of biomass, which constitutes building block metabolites (i.e. amino acids, lipids, carbohydrates, nucleotides), inorganic ions, vitamins, and co-factors [8]. Flux Balance Analysis (FBA) enables the estimation of maximal biomass production, i.e. yield or rate, depending on how the model is constrained [9]. For example, biomass production rate can be predicted by Flux Balance Analysis (FBA) if the inflow of nutrients to the metabolic network is constrained

**Funding:** CK acknowledges support by the Collaborative Research Centre 1182 - "Origin and Function of Metaorganisms" - Deutsche Forschungsgemeinschaft. CK and SW acknowledge support by the Cluster of Excellence 2167 - "Precision medicine in chronic inflammation" - Deutsche Forschungsgemeinschaft. The funders had no role in study design, data collection and analysis, decision to publish, or preparation of the manuscript.

**Competing interests:** The authors have declared that no competing interests exist.

by the maximum nutrient uptake rate [10]. However, these uptake rates are context-, organism-, and compound-specific, which render the experimental measurement of the values for many FBA applications infeasible especially in chemically complex environments (i.e. broad range of potential nutrients). In contrast, if FBA models are directly constrained by the concentrations of nutrients, the biomass production yield can be predicted in units of grams biomass per gram nutrient [9]. The accuracy of the FBA model predictions are dependent on three factors: (1) A properly defined linear objective function that accurately represents the cell's biochemical objective, (2) an accurate reconstruction of the metabolic network architecture that represents the cell's metabolic capabilities, and (3) a quantitative representation of the chemical environment with all nutrients that are available for uptake. Several approaches and methods have been developed to construct realistic biomass objective functions [11,12] and to increase the accuracy of the metabolic network structure [13–16]. A realistic representation of the nutritional environment has received far less attention, although it is of increasing importance, as flux balance analysis is progressively applied in contexts, such as environmental and clinical microbiology. For example, it is challenging to study the nutritional microenvironment of bacterial cells in the mammal's intestine. In such cases, the exact chemical composition of the environment can only be vaguely defined, as the individual's diet is a source of variation [17] and the bioavailability of nutrients (especially important for the bioactive ones) is dependent on various processes inside the intestine [18].

Thus, it is a common issue that there is no unanimous approach on how to model the nutritional input in constraint-based modelling. For instance, it is usually not well-defined how to proceed in representing various compounds (e.g. macromolecules such as proteins, isomeric forms of compounds such as L- and D-Lactate, volatile compounds, gases, and trace elements) in the models and how to proceed in calculating their inflow values (i.e. constraints) starting from their concentrations. Researchers often unintentionally fail to provide information on their exact methodology how nutritional constraints were calculated and in numerous instances arbitrary numbers are used for nutritional constraints [19]. As a result, the validity of phenotypic predictions is potentially impaired and the applicability of FBA models therefore limited. Thus, new methods and guidelines are required to increase the predictive potential of FBA models especially in chemically complex environments and to facilitate reproducibility of results obtained from computational models of metabolism.

Here, we provide clear guidelines on how to design a realistic computational representation of the nutritional input for metabolic models for environments, which are initially only defined on the level of chemically complex components such as yeast extract, tryptone or milk powder. Therefore, a step-by-step procedure is proposed that incorporates publicly available databases and resources and facilitates reproducible calculations from chemically complex components to the quantities of specific molecules that are presumably available for uptake by the cell types of interest. The procedure is illustrated on the example of Lysogeny Broth (LB) medium, a commonly used microbial growth medium (experimentally and *in-silico*) that contains, besides water and NaCl, the chemically complex components tryptone and yeast extract which provide a broad range of nutrients for microbial growth. Based on the example we emphasise the importance for rationally designed nutritional constraints in FBA models in order to obtain meaningful physiological predictions.

## Material and methods

### Software for data analysis and metabolic modelling

Nutritional data was stored and processed using R (v. 3.6.3, [20]) and the R-package 'readODS' (v. 1.6.7, [21]). FBA and further model analyses were performed using the R-packages

'glpkAPI' (v. 1.3.2, [22]), 'sybil' (v. 2.1.5, [23]), 'sybilSBML' (v. 3.0.1, [23]), and the software libSBML (v. 5.17.2 [24]) and GLPK (v. 4.65 [25]). For simulating the metabolic network of *Escherichia coli* str. K-12 substr. MG1655, we employed the model 'iJO1366' [26], which can be downloaded from the BiGG database (http://bigg.ucsd.edu/) [27,28]. The model's default, "core" biomass reaction with an ATP-consumption estimate for growth associated mainte-nance (GAM) of 53.95 ATP gDW$^{-1}$ was defined as an objective function. For data processing and visualisation the R-packages 'dplyr' (v. 0.8.5, [29]) and 'ggplot2' (v. 3.3.0, [30]), 'egg' (v.0.4.5, [31]) were utilised.

## Stepwise procedures

The methodology is graphically depicted in Fig 1. In the following section, we are going to describe the modelling procedure, which is subdivided into 6 distinct steps. To illustrate the proposed procedure, this protocol is applied to computationally model the media conditions of Lysogeny Broth (LB) for cultures of *Escherichia coli* str. K-12 substr. MG1655 (S1 File: Step-wise procedure of designing nutritional input (LB medium)).

## Step 1: Understanding the chemical composition of the growth medium and the physiology of the model organisms

The first task of nutritional modelling is the collection of information regarding the growth medium of the organism. One of the aims of this initial step is to gather information about the

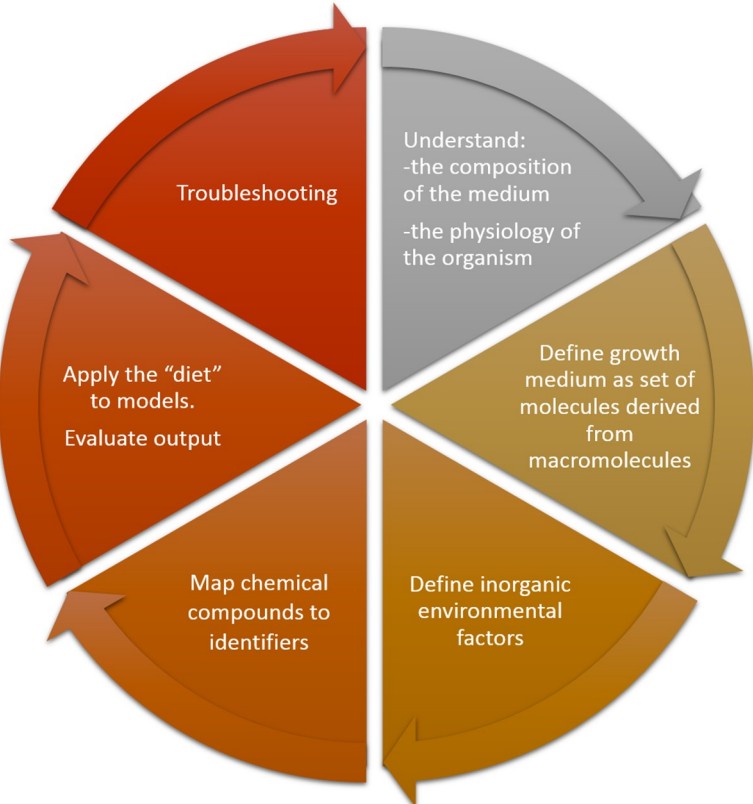

**Fig 1. Graphical summary of stepwise procedures.** Roadmap to definition of the nutritional input for genome-scale metabolic models.

**Table 1. Examples of nutritional model for bacteria, humans and worms.**

| Nutritional model or medium | Source |
|---|---|
| Defined culture medium for *Escherichia coli* | *Brock Biology of Microorganisms*, p. 100, ISBN: 978-1-292-01831-7 [33] |
| Various human diets (e.g. Vegetarian, Mediterranean) | www.vmh.life [34,35] |
| *Caenorhabditis elegans* Maintenance Medium | Szewczyk NJ *et al.* [36] |

background of the organism of interest (e.g. metabolic functions, auxotrophies). Additionally, a further aim is the collection of data that will enable us to enumerate and quantify those components that can potentially serve as nutrients for the model of this organism.

In laboratory settings, this information can usually be extracted from the definition of the culture medium. For that purpose, the modelling of qualitatively and quantitatively detailed defined growth media can be straightforward, as direct measurement/definitions of specific molecules (e.g. measurement of amino acids in a liquid [32]) might be provided. On the other hand, in environmental settings (e.g. soil, sea water, mammalian intestine), data on the chemical composition of the growth environment might be less accessible but could potentially be gathered from online databases and available literature (see Tables 1 and 2 for example resources).

Furthermore, modelling the metabolism of a cell requires detailed knowledge of the underlying (micro)environment, because cells are dependent on nutrients provided either through the medium or from other neighbouring cells. Therefore, they cannot be regarded as isolated organisms *per se*.

For instance, the simulation of cells, which are part of a multicellular system (e.g. host cells and bacteria in the vicinity [47]), requires more elaborate equations that link the molecular

**Table 2. Collection of the most useful resources for modelling nutrition.**

| Database | Website / ISBN of the book | Short Description |
|---|---|---|
| Metabolic modelling databases | | |
| VMH | www.vmh.life [34,35] | Metabolic models of bacteria, human cells, and collection of human diets |
| BiGG | https://bigg.ucsd.edu [27,28] | Metabolic models of various organisms |
| ModelSeed | www.modelseed.org [37] | Construction of metabolic models with collection of reactions/metabolites |
| MetaNetX | www.metanetx.org [38–41] | Integration of different metabolic modelling databases |
| Chemical and nutritional databases | | |
| PubChem | https://pubchem.ncbi.nlm.nih.gov [42] | Chemical compounds and their properties |
| FoodData Central | https://fdc.nal.usda.gov [43] | Chemical composition of ready and raw foods. |
| | | Until 2019 the database was available through USDA Food Composition Databases (https://ndb.nal.usda.gov/ndb/) |
| FooDB | www.foodb.ca/ [44] | Chemical and biological database specialised on foods and nutrients. |
| Fundamental nutritional literature | | |
| Ernährung des Menschen | ISBN 9783825287481 [45] | Detailed overview of human nutrition (in German) |
| Advanced Nutrition and Human Metabolism | ISBN 9781305627857 [46] | Detailed overview of human nutrition and metabolism (in English) |

quantities of compounds with what is available for a model as input. For example, to simulate the nutritional environment of bacterial cells in the lower gastroenterological tract, we should account for the chemical and biological processes that take place in the upper tract, as host processes contribute to the bacterial microenvironment. Such processes may include absorption [for details see [48]], enzymatic and spontaneous degradation or modification of chemical compounds, and other time-dependent ones (e.g. peristalsis).

On the other hand, bacterial cell cultures involve well-designed experiments and many parameters are known and/or adjusted according to the scientific question (e.g. chemically defined media, defined environmental conditions). In such cases, this information can be directly incorporated into *in-silico* experiments. In conclusion, modelling should be firstly focused on clarifying the nutritional environment of an organism.

Here, we will consider the Lysogeny Broth (LB) medium as a sample case [49]. This medium is considered complex and nutrient-rich and is widely used for the cultivation of a range of different bacterial species [50]. It is an aqueous mixture composed of tryptone, yeast extract, and sodium chloride, while sodium hydroxide is added to adjust the pH level to 7.0 [49]. Although production of yeast extract is not unanimous and may differ across different products, in this study, we assumed that the yeast extract (*Saccharomyces cerevisiae*) is autolysed (i.e. lysis of yeast contents by the cell's own enzymes) [51].

Tryptone and yeast extract are the major constituents of this growth medium. The chemical composition of those ingredients can often be obtained directly from the manufacturer's product specifications. Such manuals ideally contain detailed nutritional information about the major elements and amino acids, which are available in yeast extract and tryptone. However, these manuals typically lack information about the content of minor minerals, vitamins, nucleic acids, fatty acids, and fibres. These constituents were for example inferred from the studies by Grant *et al.* [52], by Sarwar *et al.* [53] and as well as by the *Bionumbers* [54,55]— along with the bionutrients technical manual published by the company BD [56].

## Step 2: Define growth medium as a set of molecules derived from macromolecules

At this stage, the objective is the establishment of a list of molecules and their quantities based on the known ingredients of the medium (see step 1). Such a representation of the composition of the growth medium has to be derived, because only a small set of compounds of the medium are represented in the models.

Nutrients in organismal metabolic models are usually depicted as unique molecules with known chemical formula, structure, and mass such as vitamins, ions, amino acids, fatty acids, mono-/oligosaccharides, and specific fibres. By using primary literature, textbook information, and databases, (see Table 2 for example resources), it is possible to translate macromolecules or mixture of molecules into their constituent molecular components (e.g. albumin into its amino acid composition or mineral mixes to individual vitamins and minerals, see Table 2 for examples).

After collecting all the compounds and their quantities, unit transformations to the reference unit of the study (e.g. mmol/L) should be performed. Usually, chemical compounds are dissociated or hydrated in aqueous solutions (e.g. salts) or are dependent on the pH of the solution (e.g. conjugate bases and acids). Such cases ought to ideally be represented by one molecule or ion (e.g. butyrate not both butyrate and butyric acid), since models can often only handle one form of molecules or ions. A special case are natural polymers (e.g. fibres), which can occur in many different configurations and lengths. However, metabolic models often comprise only specific forms of the polymers. In such cases, one solution is to assume that the

focal polymer occurs only in the forms that the model supports. Another possibility to deal with the multiplicity of polymer variants is to translate the total polymer mass within the growth medium to a concentration of the respective monomers. However, in this case, the model would not require the enzymatic hydrolysis of the polymer.

Commercial mixture of compounds (e.g. yeast extract, casitone) can be translated into their molecular components (e.g. ions, amino acids) according to the manufacturer's manuals. To sum up, the focus of this step is the enumeration of compounds which constitute the growth medium of interest. In our example, we followed these principles by adding ions of salts (e.g. NaCl recorded as ion Na+ and ion Cl-) or simple compounds (e.g. polymer cellulose recorded as its dimer cellobiose).

## Step 3: Define the inorganic environmental factors

The objective of this step is to define the environmental compounds needed for cell growth. This step is important, because there are environmental constraints of the actual experimental or physiological setting that should be taken into consideration in addition to the nutritional compounds. For example, culturing conditions are usually well defined with respect to the oxygen regime as well as pH. Thus, the concentrations of the substances oxygen and protons need to be adapted accordingly (i.e. [oxygen] = 0 mM for anaerobic cultures, [protons] = $10^{-7}$ M for medium pH 7).

In our illustrative example (LB medium), the original experiments are conducted in aerobic conditions in an aqueous solution. Assuming that water's density is 1 g/cm$^3$, concentration of water in pure water is $5.55^*10^4$ mmol/L and the concentration of oxygen was set to 18.2 mmol/L, based on data from *E. coli* cultures [57]. Concentration of protons in the computational diet should reflect the pH of the actual experiments (pH = 7 which is equal to $10^{-7}$ M or $10^{-4}$ mM of protons) [49].

## Step 4: Map chemical compounds to metabolite identifiers of the metabolic model

The goal of this step is to map molecules to their representations in the model. It is a crucial step of metabolic modelling, as the more precisely the list of compounds is matched to the metabolites of the model, the more accurate will be the outcome of model simulations. In general, metabolites are linked to specific exchange reactions, i.e. pseudo-reactions that enable the exchange of metabolites with the external compartment of the model, which represents the theoretical chemical environment of the model.

In the general convention for metabolic modelling, exchange reactions that represent the chemical environment have identifiers with prefix "EX_" followed by the identifier of the focal metabolite/nutrient and the suffix for the external compartment (i.e. "(e)" or "_e"). However, the exact nomenclature for exchange reactions depends on the underlying reaction database and modelling framework. Although databases are not always compatible, approaches for mapping identifiers are available (e.g. MetaNetX [38–41]). Nonetheless, some databases such as VMH [34,35] and BIGG [27,28] share some but not all metabolite and reaction identifiers. The most commonly used biochemistry databases for metabolic modelling can be found in Table 2.

Furthermore, compounds, which are part of the growth medium, are frequently not available in a model (e.g. fibres). There are three options available to address this issue: (1) to remove these compounds from the setting (e.g. dyes and indicators of *in-vitro* cultures like phenolphthalein can be excluded if we can assume that they do influence the organism's metabolic processes), (2) choice of another molecule which is related to the missing compound (e.g.

nicotinate instead of nicotinamide), or (3) if feasible, to curate the models, i.e. to extend the models by adding the exchange and internal metabolic reactions needed based on the available genetic or physiological evidence. All in all, the objective of this step is the most precise representation of the molecular composition of the chemical environment in the model, as incorrect assignment of metabolite(s) to model compounds frequently leads to errors in subsequent model applications (see troubleshooting step).

Our LB practical example is based on the iJO1366 [26] model of *E. coli*, which can be found in the BIGG [27] database of models, reactions, and compounds. In total, 53 compounds were found in the metabolite list of the model. For 13 cases we could not identify appropriate exchange reactions in the model (e.g. metals, folate), so they were not added in the setting. Moreover, some media compounds are matched with more than one model metabolite. For instance, concentration of ions in media definitions are frequently provided without specifications of their oxidation states (e.g. $Fe^{2+}$ / $Fe^{3+}$), while metabolic models require this information as the redox potential is crucial for most reactions that involve these ions. A quantitative estimation of the oxidation states ratio of ions in rich media such as LB will be in most cases not possible without experimental measurements, as such ratios are highly variable over time e.g. by reacting with dissolved oxygen. In our LB medium example and in the case of Fe ions, we have chosen a 50% $Fe^{2+}$ / 50% $Fe^{3+}$, which reflects the measured intracellular ratio of the oxidation states in *E. coli* W3110 [58]. For chemically well-defined media formulations, the concentration of the different oxidation states can be inferred from the amount of the source salt (e.g. $FeSO_4$ provides $Fe^{2+}$, $FeCl_3$ provides $Fe^{3+}$ when dissolved).

## Step 5: Apply the "diet" to the model and evaluate the output

This step focuses on adjusting the exchange reactions linked to the metabolite identifiers, which were selected in step 5, and on performing growth simulations. This adjustment is essential for the accuracy of the growth simulation, as the outcome (i.e. growth) depends on the availability of each metabolite according to the actual concentration of the diet.

Notably and in the framework of FBA, the flux of the exchange reactions should be constrained to represent the availability of the nutrients. For this purpose, exchange reactions are constrained by upper and lower bounds. Usually, the uptake of a compound is represented by a negative value and excretion by a positive value. In case the FBA modelling approach aims to predict the organism's growth rate ($hr^{-1}$), the concentrations defined after step 1 to 4 need to be transformed to the maximum uptake fluxes in the unit $mmol*(gDW*hr)^{-1}$. This conversion is not trivial as it relies on kinetics (e.g. following the Monod equation), which are dependent on the compound and the organism of interest. In case the aim is to predict maximum growth/biomass yield (gDW per litre of growth medium), one can assign the concentration of the nutrients (e.g. in mmol per litre of growth medium) to the uptake bound as explained in the introduction.

Having defined constraints on the exchange reactions, the models can be optimised typically by maximising flux through the biomass reactions. Usually software like sybil [23], which utilises FBA [9], or the various COBRA suites/tools [59–62] are useful for this purpose. If the input of the exchange reactions reflects the concentration of compounds, the result of the optimisation corresponds to the biomass yield based on the growth medium. Otherwise, if the input reflects fluxes, the result corresponds to the actual growth rate of the organism.

In our LB example, the models followed the nomenclature scheme "EX_" (for exchange) + metabolite identifier +"(e)" (for external) to represent exchange reactions. In our case, by utilising sybil [23], the exchange reactions are constrained based on the concentration of the growth medium and used to conduct FBA. For compounds not present in the diet we set the lower bound to zero.

## Step 6: Troubleshooting

The aim of this step is to reveal and solve issues that occur after applying the computational diet and optimisation methods (i.e. FBA). Troubleshooting and fine-tuning should ideally guarantee that the outcome is biologically meaningful.

For instance, although it might be known that the cells can grow on specific medium, the simulation may lead to an unexpected outcome, where the model does not grow after following all the above steps. Except for the case of an erroneously defined diet, this issue may be associated with low quality of the model, which may miss reactions or the model may have gaps and/or errors. If the latter is the case, then either the model needs to be refined (model curation) and gap-filling algorithms can be applied [63,64].

In case of an erroneously defined nutritional growth environment, reduced cost calculations of uptake reactions can provide insights on nutrients that are absent or have too low concentrations in the modelled medium but which are essential for cell growth [65]. Reduced costs are sensitivity parameters for each reaction that are part of an FBA solution. Briefly, these values can be interpreted as the impact of an increase of the flux through the respective reaction on the final objective value (i.e. growth yield) [66]. Thus, when applied to the nutrient uptake reactions, the reduced costs allow the identification of compounds that could be added to the media in order to obtain increased or enabled growth [67].

Therefore, reduced costs can indicate compounds, which are part of the actual nutritional environment, but are not included in the modelled computational growth medium due to missing chemical data and knowledge of the media composition. In such cases, steps 1–5 have to be re-evaluated potentially by including additional metabolomic/chemical information on the medium components from the literature or even performing targeted chemical analytics to quantify the metabolite in question. In addition, the reduced cost analysis could also point to compounds, which may already be part of the modelled nutritional environment, but whose quantity may not be sufficient to facilitate model growth. This could be for instance the case if the lower bound for ATP hydrolysis, which is often part of cellular models to represent energetic costs for cell maintenance, cannot be reached due to too low availability of potential energy sources.

In this context, it also needs to be noted that a zero or infeasible growth solution returned by the FBA model under the reconstructed nutritional environment does not necessarily indicate missing or limiting compounds in the defined diet. Instead, a *nil* growth solution could also be caused by inconsistencies in the model itself, i.e. due to missing reactions, incorrect reaction bounds, or erroneous definition of the biomass reaction. Thus, in case of *nil* growth it is recommended to consider also options for model curation and algorithmic gapfilling approaches (e.g. [13,63,64]). Alterations to the modelled nutritional environment in order to enable growth should be primarily considered in cases where the model itself underwent already substantial curation efforts and where uncertainty of the exact chemical composition of the growth environment exists. The latter is especially relevant for growth media, which are generally chemically undefined such as LB medium or other complex growth media that are frequently used for animal/human cell line cultures [68].

## Results

### *E. coli* growth in modelled LB-medium

As an exemplary case, we applied the proposed procedure to the chemically complex growth medium Lysogeny Broth (LB) [49]. Following step 5, the optimal growth yield of the model of interest based on the user-defined nutritional constraints can be calculated. The modelled LB

medium was applied to constrain the genome-scale metabolic model of Escherichia coli K-12 (model designation: iJO1366). At first, the *E. coli* model predicted that the organism is not able to produce biomass in LB medium (objective value of zero). To troubleshoot this issue, we searched for the reduced costs. The missing compound with the highest absolute reduced cost was molybdate, EX_mobd(e), a molecule comprising molybdenum and oxygen. Ions of molybdate are the only biologically relevant source of molybdenum and play an important role in redox enzymes [69]. Molybdenum occurs in a variety of food categories [70]. The concentration of molybdate can be inferred by the value of molybdenum in this case. Adding molybdate (lower bound -3.07*$10^{-4}$; value as molybdenum in STEP 3, based on the initial value from [52]), the optimisation was possible and the value of objective function was 1.54 g per Litre of LB medium for iJO1366. This case illustrates that calculating the reduced costs can indicate essential compounds that are missing in the nutritional constraints of the model and to adjust die diet composition accordingly.

In addition, we applied the modelled LB medium (incl. molybdate) to the core model of *E. coli* metabolism, which is a sub-network of the *Escherichia coli* K-12 genome-scale metabolic network [71] and which can be downloaded from the BiGG database (http://bigg.ucsd.edu/) [27,28]. The predicted biomass yield using this core model reached 0.59 g/L, which is only 38% of the growth yield predicted using the genome-scale model. This decrease is explained by the fact that the core model does not include the reactions and transporters that are required to utilise a number of nutrients from the LB medium, which the genome-scale model is able to use for growth.

*E. coli* is known to produce acetate as metabolic by-product at high growth rates, even in the presence of oxygen as potential electron acceptor [72]. Using our *in silico* representation of the LB medium, we also predicted acetate production using the core model (4.5 mmol/L yield) and the genome-scale model (46.72 mmol/L).

## Impact of diet modifications

Based on the rationale that an accurately modelled growth medium or diet allows realistic prediction of the cellular metabolism and cell growth, those models should also be able to accurately predict the metabolic- and growth response to medium modifications, such as nutrient supplementations. To test this, we searched for dietary compounds that increase the predicted objective function value (i.e. *E. coli* growth yield) if added as a supplement to the computational LB medium. To this end, we repeated the optimisation by adding +1mM (case A) or 30% (case B) of all compounds contained in the LB medium. We used these two different supplementation approaches to obtain substantial (cut-off of relative difference = $10^{-6}$) effects on growth both when supplementing compounds which are initially low in abundance or absent or whose molecular weight is high (case A) and when supplementing initially highly abundant compounds or low-molecular weight compounds (case B). Additionally, we have extended the list of supplemented compounds (case A) to the carbohydrates D-glucose and L-arabinose, as glucose is very often used in growth experiments and arabinose has been identified in yeast extracts in a previous publication of Sarwar *et al.* (53). The analysis revealed that most compounds, which increased the growth of the model, mainly belonged to the group of amino acids, and secondarily to nucleobases, carbohydrates, and fatty acids (Fig 2A). Oxygen had the strongest effect on biomass yield (more than 17% increase of yield while supplementing +30%; see Fig 2B).

Moreover, supplementation of carbohydrates had the strongest effect on the predicted yield compared to the other nutrient groups (Fig 2A). This suggests that the available amount of carbohydrates in LB medium is growth yield limiting. Interestingly, the effect of supplementing

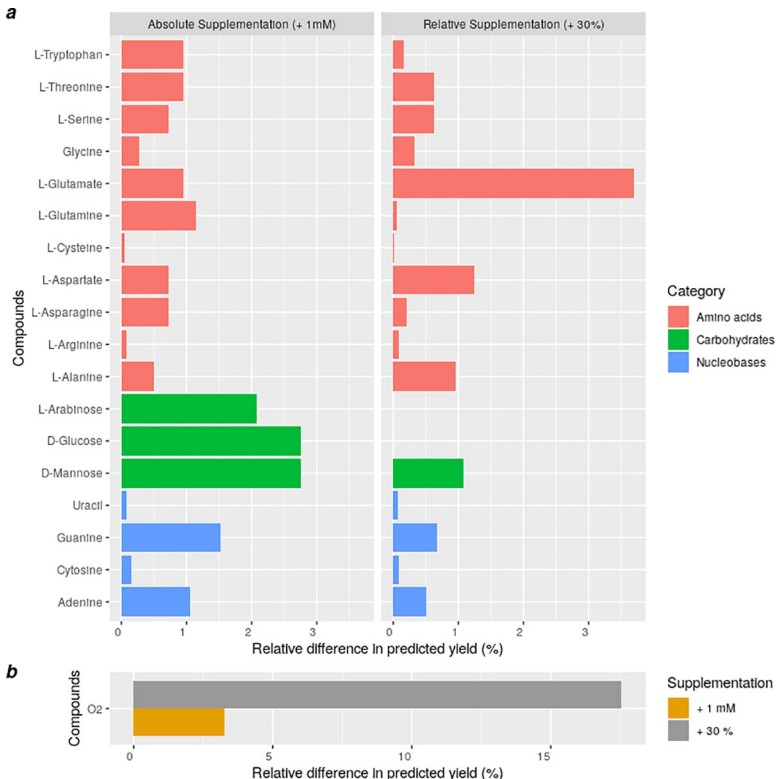

**Fig 2. Impact of diet modifications.** (a) Supplementation of amino acids and nucleobases to the computational LB medium (+30% of the original amount or +1 mM for each compound) and relative differences to the optimisation results for growth yield of the model iJO1366. (b) Similarly to (a) for oxygen. Cut-off of relative difference = $10^{-6}$.

the purines guanine and adenine is substantially bigger in comparison to the effect for uracil and cytosine, possibly due to the difference in the initial concentrations (mean concentration for G/A ~0.32mM, for U/C ~0.24mM). In contrast, the supplementation of arginine, cysteine, and glycine is predicted to have only limited effects on the growth yield (Fig 2A).

## Discussion

### Impact of defining precise nutrient levels for metabolic models

The aim of this work is to provide guidance for estimating the molecular concentrations of nutrients, which can be used as nutritional input in metabolic models (Fig 2). The utilisation of generic and arbitrary values can have a strong impact on downstream consequences on model predictions. For instance, recently published western and high-fibre diets, used for modelling the nutritional environment of the gut microbiome [19], contain values of 1 mmol/ (gDW*h) for a big set of exchange reactions in the section "Minerals, vitamins, other", suggesting that there is uncertainty regarding the values that should be used. Another similar example is the LB medium, which is stored in the KBase database which is a popular tool for automated model reconstructions [73]. In this case, all fluxes are 100 mol/(gDW*h) for all compounds, which is physiologically unrealistic and, thus, could result in unlikely growth and metabolic flux predictions. Additionally, automated gap-filling on inaccurate media as part of the model construction could not only cause wrong simulation outcomes (fluxes / growth), but also can cause errors in the network structure itself by making models reliant on an unrealistically high

inflow of specific compounds. As a result, such models may also require unrealistic nutrient input for growth.

As a result, a crucial question arises: Is it worth spending effort and resources in designing a computational medium with accurate quantification? To answer this question, we constrained a publicly available and manually curated *Escherichia coli* K-12 model [26] with the LB medium, which was modelled using the above described 6-step procedure and scrutinised the FBA model prediction with experimental data (growth yield, nutrient utilisation, and the effect of nutrient supplementation) of the physiology of *E. coli* cultures in LB medium.

The growth patterns of *E. coli* in general have been extensively studied [74]. For instance, it is known that the optical density of the cultures in LB Medium can achieve a value of $OD600_{nm}$ 5 [75] and up to $OD600_{nm}$ 6.49 [49]. Additionally, it has been empirically estimated that an optical density of $OD600_{nm}$ 1 corresponds to ~0.3 g/L of bacterial dry weight [55,76]. By combining the information, a yield of (5*0.3) 1.5 g/L can be calculated for the phase when growth curves reach a plateau, which is in line with our optimisation result (1.54 g/L). Interestingly, we observed acetate as metabolic by-product alongside the formation of biomass, which is in line with experimental studies that report acetate production under aerobic conditions in LB medium [72]. Taking into account these results and based on the modelled diet for LB medium with rationally defined quantities for all included compounds, it can be suggested that an accurate representation of the chemical composition of the environment is crucial for precise predictions of biomass formation and metabolic functions using FBA models.

Such realistic results can be particularly helpful for making context-specific predictions. For instance, it is possible to investigate the effect of manipulating the nutritional composition on the predicted phenotypes. In Fig 2A we presented the relative difference of the predicted yields of the *E. coli* model iJO1366 after supplementing the computational medium with each compound contained in LB medium. The analysis highlighted that glucose can increase the yield of the model, which is in line with the fact that glucose can boost the growth of *E. coli* cultures [78]. It is worth mentioning that by adding 20 mM of glucose to the simulations of the *E. coli* model, the predicted yield increased by 54%, which is an overestimation compared to the published experimental value of 39% [77]. Nevertheless, the simulations *in-silico* reproduced the strong growth-supporting effects of glucose on *E. coli* growth. Additionally, supplementing the computational diet by adding 1mM of amino acids significantly supports the growth of the model following this order: tryptophan, threonine, glutamate, serine, aspartate, alanine, glycine, while supplementing the diet by adding +30% by this order: glutamate, aspartate, alanine, threonine, serine, glycine, tryptophan. Similarly, *E. coli* has been reported to sequentially consume these amino acids in the order of serine, aspartate, tryptophan, glutamate, glycine, threonine, and alanine in a tryptone medium [49,78]. This suggests the order of consumption might be associated with the growth benefits for the cell. Taking everything into account, such predictions may be helpful for optimising cultures *in-vitro* by targeted modifications of the medium composition [79].

## Quality of FBA model prediction depend on the accuracy of the modelled environment

Although Flux Balance Analysis can be used to make useful physiological predictions, these are not limitation-free. FBA-based predictions are especially influenced by the exact values for the nutrient uptake reaction constraints. For instance, in the above-mentioned example of *E. coli* growth in LB medium, we showed that increasing the bound of oxygen uptake by 30% results in more than 17% increased growth yield. The amount of oxygen in our simulations is based on the uptake rate observed in experiments [57] and not on the environmental amount

of oxygen, because various phenomena including the diffusion from air to the medium [80] and the uptake limits by the cell [81] restrict the available amount of oxygen. Moreover it is sometimes laborious to identify such phenomena and to include them in the static models, but also it can be challenging to simply quantify some environmental constraints (e.g. gases) or variability of the nutritional compounds in the media, as discussed above, although the output of the models severely depend on them (e.g. oxygen, yeast amino acids). As a result, it is worth noting that as long as an accurate modelled environment cannot be designed, the FBA predictions will be prone to shortcomings.

## Relevance of nutritional modelling for human microbiome research

Computational predictions based on constraint-based models of microbial metabolism have emerged as a powerful tool in microbiome research and broadened its horizons [5,19,82,83]. The microbiota, especially within the human gut, has been acknowledged to be a substantial contributing factor in health and disease of the host [47]. Since the models are utilised in exploring health/disease-related bacteria-bacteria and host-bacteria interactions, the importance of diets has been recognised [84]. For instance, the predictions derived from testing various "diets" in metabolic models led to successful dietary interventions in humans [85].

By providing the community with a roadmap and a detailed example we set the basis for the development of standard operating procedures for reconstructing the nutritional environment in constraint-based metabolic modelling. The procedure for *in-silico* modelling of the nutritional environment of cells described in this study may allow efficient recognition, tracking, and handling of inaccuracies and artefacts in constraint-based metabolic modelling approaches.

## Limitations and outlook

We introduced and applied the proposed procedure to reconstruct the nutritional environment of metabolic models in the context of Flux Balance Analysis. Furthermore, the 6-step procedure can be analogously applied to other metabolic modelling frameworks such as Ordinary Differential Equation (ODE)—based kinetic models or Elementary Mode (EM) analysis; with the only exception of the proposed reduced costs analysis in step 6 ('troubleshooting'). This analysis is limited to FBA applications, as reduced costs are sensitivity parameters that are directly associated with an FBA solution [85].

A precise definition of chemical growth environment is especially valuable for kinetic modelling, as such models typically allow more quantitative predictions than FBA simulations [86]. These predictions include for example the temporal-dynamic changes of individual metabolite levels, including metabolic by-products. However, kinetic models are usually limited to small or medium-scale networks that comprise fewer reactions than genome-scale metabolic network reconstructions. Thus, kinetic models may not contain all pathways in which exogenous compounds from a complex nutritional environment enter the cell's metabolism. For the example of *E. coli*, kinetic models are often derived from a so-called *core model*, which is a sub-network of the *E. coli* genome-scale model that comprises glycolysis, the pentose phosphate pathway, the TCA-cycle, and the oxidative respiration chain [87]. This sub-network does not include the uptake and utilisation of a majority of compounds that are part of complex growth media used for the cultivation of *E. coli* such as LB medium and that are known to be utilised by *E. coli* (e.g. most proteinogenic amino acids). Hence, kinetic models on the basis of this core network cannot reproduce the metabolism of *E. coli* in complex nutritional environments. We therefore emphasise that future research should focus on the extension of kinetic models that include the uptake and degradation of several nutrients found in complex

media. In combination with the incorporation of regulatory circuits such as catabolite repression [88], kinetic modelling will highly benefit from the here proposed procedure to model complex growth environments.

Growth media for eukaryotic cell cultures are also often molecularly complex due to components such as bovine serum or meat extract, which provide a wide range of potential nutrients for cell metabolism [68]. Thus, computational approaches to simulate metabolic processes in eukaryotic cell cultures (e.g. human/animal cell lines) also depend on an accurate representation of the chemical composition of the environment. In this context, the here proposed procedure to model complex nutritional environments will also promote the predictive potential of *in silico* models of eukaryotic cell metabolism.

## Supporting information

**S1 File. Diet table.** Stepwise procedure of designing nutritional input (LB medium). (ODS)

## Acknowledgments

The authors thank Johannes Zimmermann for helpful discussions.

## Author Contributions

**Conceptualization:** Georgios Marinos, Silvio Waschina.

**Data curation:** Georgios Marinos.

**Formal analysis:** Georgios Marinos.

**Funding acquisition:** Christoph Kaleta.

**Investigation:** Georgios Marinos, Silvio Waschina.

**Methodology:** Georgios Marinos, Silvio Waschina.

**Project administration:** Georgios Marinos, Silvio Waschina.

**Resources:** Christoph Kaleta.

**Software:** Georgios Marinos.

**Supervision:** Christoph Kaleta, Silvio Waschina.

**Validation:** Georgios Marinos.

**Visualization:** Georgios Marinos.

**Writing – original draft:** Georgios Marinos.

**Writing – review & editing:** Christoph Kaleta, Silvio Waschina.

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
