## [Decision Letter · Decision Letter 0]

29 May 2020

PONE-D-20-13622

Defining the nutritional input for genome-scale metabolic models: a roadmap

PLOS ONE

Dear Dr. Waschina,

Thank you for submitting your manuscript to PLOS ONE. After careful consideration, we feel that it has merit but does not fully meet PLOS ONE’s publication criteria as it currently stands. Therefore, we invite you to submit a revised version of the manuscript that addresses the points raised during the review process.

The Reviewers acknowledged the merit of the manuscript but raised some general concerns about the generality of the approach. All specific observations should be addressed with care because I believe they would improve considerably the scope of the method. For now my decision is major revision.

We look forward to receiving your revised manuscript.

Kind regards,

Andrea Pagnani, Ph.D

Academic Editor

PLOS ONE

Reviewers' comments:

Reviewer's Responses to Questions

**Comments to the Author**

1. Is the manuscript technically sound, and do the data support the conclusions?

Reviewer #1: Yes

Reviewer #2: Yes

2. Has the statistical analysis been performed appropriately and rigorously? 

Reviewer #1: Yes

Reviewer #2: Yes

3. Have the authors made all data underlying the findings in their manuscript fully available?

Reviewer #1: Yes

Reviewer #2: Yes

4. Is the manuscript presented in an intelligible fashion and written in standard English?

Reviewer #1: Yes

Reviewer #2: Yes

5. Review Comments to the Author

Reviewer #1: The paper is a systematic review and guide to correctly setting up the environment (through exchange fluxes) in genome scale models of metabolic networks, with a clear focus on FBA or FBA-like models. In my view the article is clear (didactic) and comprehensively covers the topic. The presentation is completely consistent with the scope of the paper and could indeed be useful for many applications of FBA in biotechnology. I therefore have no major criticism to raise. I would however ask authors to address the following points, all of which are in my view important (despite being related to one another).

1) In the example they use throughout the paper (E coli growth on LB), the fact that something is wrong is reflected in the growth rate being nil (we know that E coli grows on LB). I understand that by adding a specific compound to the medium one cures this problem. My guess however is that one could cure this problem in a few other ways (eg reaction reversibilities, bounds on ATP consumption etc). My question is: when are exchange fluxes the most reasonable variables to adjust in such cases? This point somehow lies on top of the issues discussed in the paper but I believe it would be important to address it.

2) Besides adding compounds, one could solve the zero growth problem by removing and/or replacing compounds from the medium or by changing the availability of existing compounds. For instance, because there is a lower bound on ATP hydrolysis, growth in FBA is impaired when the max glucose uptake is too low. So adding compounds does not seem like a general recipe to adjust the medium composition. The authors should address this point.

3) I understand the review is focused on FBA-like models. Still in principle some of the ideas presented here could be important in other theoretical frameworks (eg network expansion, dynamical models etc). The authors should try to highlight which parts of their recipes are general (model-independent) and which are instead specifically thought for FBA.

Reviewer #2: Please see attached document. 

6. PLOS authors have the option to publish the peer review history of their article (what does this mean?). If published, this will include your full peer review and any attached files.

Reviewer #1: No

Reviewer #2: No

---

## [Author Response · Author response to Decision Letter 0]

14 Jul 2020

Response to Reviewers:

REVIEWER #1

"The paper is a systematic review and guide to correctly setting up the environment (through exchange fluxes) in genome scale models of metabolic networks, with a clear focus on FBA or FBA-like models. In my view the article is clear (didactic) and comprehensively covers the topic. The presentation is completely consistent with the scope of the paper and could indeed be useful for many applications of FBA in biotechnology. I therefore have no major criticism to raise. I would however ask authors to address the following points, all of which are in my view important (despite being related to one another).

1) In the example they use throughout the paper (E coli growth on LB), the fact that something is wrong is reflected in the growth rate being nil (we know that E coli grows on LB). I understand that by adding a specific compound to the medium one cures this problem. My guess however is that one could cure this problem in a few other ways (eg reaction reversibilities, bounds on ATP consumption etc). My question is: when are exchange fluxes the most reasonable variables to adjust in such cases? This point somehow lies on top of the issues discussed in the paper but I believe it would be important to address it."

=> We thank the reviewer for this comment and agree that adding compounds to the modelled media should not be considered as universal solution to facilitate growth of a model. We argue that this approach to achieve a desired model property (e.g. growth) should be considered as option in cases where (i) the metabolic network model underwent prior substantial curation efforts and (ii) where uncertainty exists, if a specific compound is part of the focal growth medium (e.g. molybdate in LB medium). We have added these points to the manuscript in the section for step 6 (‘troubleshooting’) on lines 332-344. 

"2) Besides adding compounds, one could solve the zero growth problem by removing and/or replacing compounds from the medium or by changing the availability of existing compounds. For instance, because there is a lower bound on ATP hydrolysis, growth in FBA is impaired when the max glucose uptake is too low. So adding compounds does not seem like a general recipe to adjust the medium composition. The authors should address this point."

=> Yes, zero-growth can also be caused by too low maximum uptake rates. In such cases, the approach to analyse the reduced costs that are associated to the FBA solution can also point towards compounds, which are already part of the modelled medium but at too low availability. We thank the reviewer for pointing out this theoretical scenario and addressed it in the revised manuscript on lines 326-331.

"3) I understand the review is focused on FBA-like models. Still in principle some of the ideas presented here could be important in other theoretical frameworks (eg network expansion, dynamical models etc). The authors should try to highlight which parts of their recipes are general (model-independent) and which are instead specifically thought for FBA."

=> We agree that most steps in our proposed procedure to model the nutritional environment for metabolic modelling approaches are not limited to FBA-models. We have added a new subsection in the discussion about the limitations of our proposed procedure and outlook by highlighting the potential applicability especially to dynamic-kinetic models of cellular metabolism (lines 498-531).

REVIEWER #2

"In this draft, G. Marinos et al. address the problem of modeling complex, possibly undefined nutritional media for their use as input for constraint based models of metabolic networks. They propose a guideline based on 6 steps to translate computationally the nutritional information for the model and show its application to the test case of the Lysogeny broth for E. Coli, where they found it improved much more the plausibility of the FBA solution both in terms of matching experimental growth physiology and overall analysis on supplementation. Although it lacks a bit of generality, the article gives an original (to the best of my knowledge) contribution towards more rigorous applications of metabolic computational modeling and thus shall deserve publication in PLOS ONE but I would suggest the author to afford the following concerns:

• It is pointed out by the authors that their aim is to make metabolic modeling more rigorous and quantitative, but then the authors apply and discuss their framework to FBA optimal solutions based on objective functions. The FBA approach is a very useful and wide spread approach eg to provide qualitative estimates of gene KO and/or medium growth feasibility but it misses sistematically even simple phenotypic (not even quantitative) predictions. For instance even for ideal E. coli cultures in minimal M9 medium (where the problem raised by the authors shall not be present) FBA fails to account for low (Journal of Biological Chemistry 278.47 (2003): 46446-46451) and high (Microbiol. Mol. Biol. Rev. 69.1 (2005): 12-50) growth rate phenotypes (as well as intermediate in non-linear way: Appl. Environ. Microbiol. 72.2 (2006): 1164-1172). In the range of growth rate explored by the authors there shall be acetate excretion, even in rich medium (Nature 528.7580 (2015): 99-104.). Is this simple prediction verified in the authors’ framework? My general question is then why the authors limit themself to FBA? In principle their framework is more general and could be more useful in more

quantitative computational modeling approaches, from dynamical kinetic models to Metabolic Flux Analysis. Although an application to these more quantitative approaches would add substantial value to their work, I invite the authors to at least acknowledge the limitations of their approach within the context of FBA."

=>We thank the reviewer for pointing out the limitations of our proposed procedure in the context of FBA-models and the potential benefit for other metabolic modelling techniques.

Following the suggestions to check for acetate production, we re-analysed the model-predictions and can confirm that the genome-scale E. coli model (iJO1366) as well as the core model predicted acetate production under aerobic conditions and on the basis of the modelled LB-medium. We have added these results and their discussion on lines 373-376 and lines 436-438, respectively.

In addition, we extended the discussion in the revised manuscript as suggested with a new subsection for the limitations of our proposed procedure to model the nutritional environment for metabolic models and an outlook for the potential impact of the procedure for other metabolic modelling approaches, namely kinetic models of cellular metabolism (lines 498-531).

"• A related issue is: how does the approach applies to smaller networks, like the core metabolism of the model iJO1366? The latter are the networks usually used for truly quantitative approaches like metabolic flux analysis. Probably the issue is non trivial since one shall reliably estimate precursors concentrations."

=> We agree with the reviewer that smaller metabolic networks (e.g. of core metabolism) are powerful frameworks for quantitative and dynamic modelling approaches. Although FBA certainly is limited in possibilities to make quantitative predictions, we have applied our modelled LB-medium to the core model of E. coli to compare predictions with the genome-scale model (see lines 365-372). While also the core model, as mentioned above, can predict acetate production alongside cell growth on LB, it lacks the capabilities to utilize a large number of nutrients that are part of this chemically complex medium. Thus, we argue that our proposed procedure for in silico nutritional environment representation will have the most impact on small-scale networks for metabolic flux analysis techniques, which include the uptake/transport reactions and nutrient utilisation pathways for several resource compounds from complex growth media. We have added this consideration and outlook to the revised version of out manuscript (lines 506-524).

"• The authors could spend few words more in the discussion on applications to eucaryotic growth media, that are the most complex and for which the impact of their framework could be bigger."

=> The complexity of eukaryotic growth media is indeed an important topic. We added a brief discussion for the potential impact of our proposed procedure to define the nutritional environment for metabolic models of eukaryotic cell metabolism (lines 525-531). 

"• Minors

line 62: ions → inorganic ions 

line 63: prediction of biomass production → estimation of maximal biomass production 

line 71: the accuracy of FBA ...depends.... → I would add that the accuracy on FBA could depend, most crucially, on the extent that cell metabolism is well described by linear objective functions.

line 329-332 “When the nutritional input of the models is defined as molecular quantities and not as fluxes the lower bound of each exchange reaction can be filled with the respective concentration of the metabolite”. I find this unclear. How is the conversion made? That in principle depends in non trivial way on the mechanism of uptake that could be different for each compound, with different kinetic constants, etc (eg Monod equation for glucose)."

=> All points were corrected for the revised version of our manuscript. We also rephrased and clarified the mentioned sentence by elaborating on the details for the conversion of concentrations and exchange reaction bounds (lines 282-289).

---

## [Editor Report · Decision Letter 1]

16 Jul 2020

Defining the nutritional input for genome-scale metabolic models: a roadmap

PONE-D-20-13622R1

Dear Dr. Waschina,

We’re pleased to inform you that your manuscript has been judged scientifically suitable for publication and will be formally accepted for publication once it meets all outstanding technical requirements.

Kind regards,

Andrea Pagnani, Ph.D

Academic Editor

PLOS ONE

---

## [Editor Report · Acceptance letter]

3 Aug 2020

PONE-D-20-13622R1 

Defining the nutritional input for genome-scale metabolic models: a roadmap 

Dear Dr. Waschina:

I'm pleased to inform you that your manuscript has been deemed suitable for publication in PLOS ONE. Congratulations! Your manuscript is now with our production department. 

Kind regards, 

on behalf of

Professor Andrea Pagnani 

Academic Editor

PLOS ONE